# Unveiling the Pathogenesis of Psychiatric Disorders Using Network Models

**DOI:** 10.3390/genes12071101

**Published:** 2021-07-20

**Authors:** Yanning Zuo, Don Wei, Carissa Zhu, Ormina Naveed, Weizhe Hong, Xia Yang

**Affiliations:** 1Department of Biological Chemistry, David Geffen School of Medicine, University of California at Los Angeles, Los Angeles, CA 90095, USA; yzuo@mednet.ucla.edu (Y.Z.); DonghuiWei@mednet.ucla.edu (D.W.); whong@mednet.ucla.edu (W.H.); 2Department of Neurobiology, David Geffen School of Medicine, University of California at Los Angeles, Los Angeles, CA 90095, USA; 3Department of Integrative Biology and Physiology, University of California at Los Angeles, Los Angeles, CA 90095, USA; czhu721@g.ucla.edu (C.Z.); ormina@g.ucla.edu (O.N.); 4Department of Psychiatry, Semel Institute, University of California at Los Angeles, Los Angeles, CA 90095, USA; 5Brain Research Institute, University of California at Los Angeles, Los Angeles, CA 90095, USA; 6Institute for Quantitative and Computational Biosciences, University of California at Los Angeles, Los Angeles, CA 90095, USA

**Keywords:** psychiatric disorders, network modeling, integrative genomics, systems biology, multi-omics

## Abstract

Psychiatric disorders are complex brain disorders with a high degree of genetic heterogeneity, affecting millions of people worldwide. Despite advances in psychiatric genetics, the underlying pathogenic mechanisms of psychiatric disorders are still largely elusive, which impedes the development of novel rational therapies. There has been accumulating evidence suggesting that the genetics of complex disorders can be viewed through an omnigenic lens, which involves contextualizing genes in highly interconnected networks. Thus, applying network-based multi-omics integration methods could cast new light on the pathophysiology of psychiatric disorders. In this review, we first provide an overview of the recent advances in psychiatric genetics and highlight gaps in translating molecular associations into mechanistic insights. We then present an overview of network methodologies and review previous applications of network methods in the study of psychiatric disorders. Lastly, we describe the potential of such methodologies within a multi-tissue, multi-omics approach, and summarize the future directions in adopting diverse network approaches.

## 1. Introduction

Psychiatric disorders, a group of prevalent brain disorders involving complex disturbances in socio-cognitive functioning, are a leading cause of disability [1]. Surpassing cancer and cardiovascular disease, psychiatric disorders are estimated to affect 792 million people worldwide, representing 10.7% of the total population [2]. Nearly one in five adults experiences a psychiatric disorder in the U.S., with major depressive disorder being the leading cause of disability [3]. Given the lifelong morbidity and dearth of rationally designed treatments, it is imperative to understand the pathophysiology of psychiatric disorders. While numerous advances have expanded the scope of genetic analysis, how genetic risk confers pathophysiology remains largely elusive.

Key unanswered questions include: How do polygenetic contributions interact to affect molecular signaling and endophenotypes? Do different combinations of common and rare variants produce distinct manifestations of psychiatric symptoms? How do we meaningfully expand the understanding of these genetic interactions, and how do we leverage such knowledge to promote precision medicine (Figure 1a)?

One promising approach is to broaden the analysis to noncoding regulatory elements and consider their effects within the complete architecture of a functional genome [1,4]. Current analytical methods mostly focus on interpreting common and rare variants located in gene coding regions, but 93% of the disorder-associated loci identified in GWAS are located in non-coding regions of the genome [5]. However, analyzing global interrelationships between non-coding regulatory elements and rare and common variants is complicated by cell-type heterogeneity, data availability, and incomplete annotation.

In this review, we discuss recent advances in network approaches that may address this complexity. Broadly, for example, network approaches that define the superstructures of interactions and probabilistic models that localize key nodes in this structure could compress the genetic search space to the most important elements. We will firstly introduce the prevailing view on the polygenetic architecture of psychiatric disorders and review genetic studies that have linked rare and common genomic loci to different conditions. We will then discuss several challenges in dissecting the full architecture of psychiatric disorders with this view. We will present advances in multi-omics approaches and network methodologies that could address these challenges in the context of an omnigenic model. We will introduce emerging network tools that are underutilized but promising. Lastly, we will summarize potential future directions of developing network approaches.

## 2. The Scope, Characteristics, and Genetic Architecture of Psychiatric Disorders

We surveyed six common psychiatric disorders with various levels of heritability (twin heritability ranging from 0.37 to 0.85) and distinct domains of manifestation and etiology: alcohol use disorder, autism spectrum disorder (ASD), bipolar affective disorder, major depressive disorder, post-traumatic stress disorder (PTSD), and schizophrenia. Details regarding disorder characteristics, heritability, and impacts are outside the scope of this review but can be found in previous reviews, such as Sullivan and Geschwind, 2019 [1].

Over the past decade, genetic studies have linked both rare and common genomic loci to different disorders and traits [6,7,8,9,10,11,12]. One efficient way to characterize the genetic architecture of complex diseases is to search for protein-encoding rare mutations in singletons or multiplex families with extreme phenotypes, which include early onset, more severe symptoms, or fast progression of diseases [13]. Human genetic studies of extreme psychiatric phenotypes and rare syndromes involving psychiatric symptoms have revealed numerous rare variants in psychiatric disorders. These rare variants include copy number variations (CNVs) and protein-altering point mutations; particularly for schizophrenia and ASD, 159 and 136 rare variants have been identified, respectively [7,14,15,16,17]. These rare mutations occur at extremely low frequencies in the population, but each with a large effect size. One rare mutation on its own may be sufficient to cause a specific disorder, as in the Rett syndrome [18]. Rare variants often cause loss-of-function of known genes, and it is relatively easier to identify the affected pathways. For instance, synaptic function and transcriptional regulation pathways have been implicated by ASD rare genetic variants. However, rare variants can only explain a small proportion of individuals [1].

By contrast, recent GWAS studies have uncovered common genetic loci with relatively high frequencies in the population but each with small effect sizes, making common variants challenging to identify unless studied in large populations comprised of as many as 100,000 subjects [19,20]. Despite the challenges, 353 common loci have been identified for the surveyed psychiatric disorders, amongst which 270 loci are associated with schizophrenia [14]. Such findings illustrate a polygenic model in which many gene loci with small effect sizes and hub genes with moderate to large effect sizes contribute to a disorder [21] (Figure 1b).

There are several challenges in investigating the polygenic architecture and mechanisms of psychiatric disorders. First, large population samples are needed to overcome statistical hurdles to identify common variants with small effect sizes. Second, interpreting the biological roles of common variants is challenging because 93% of the common variant loci located in non-coding areas of the genome can regulate gene expression in a subtle or indirect way [5]. Third, the set of regulatory actions of a given gene is diverse and varies across cell types and developmental stages. Last, cell-state-specific sequencing and functional annotations of such non-coding areas are unavailable or inconsistent.

Multi-omics and network approaches can address these levels of complexity by exploiting the structure of physiological contextualization and connectivity to generate otherwise inaccessible insights. Multi-omics approaches integrate genetics, functional genomics, transcriptomics, proteomics, and epigenetics. Integrating multiple levels of analysis in this manner can provide unique windows into key driving elements and hypothesized biological functions of genes that would otherwise be opaque to a single-level analysis. One way to integrate multi-omics data is through network approaches and we will discuss the biological and pragmatic motivations for transitioning from a polygenic model to an omnigenic network model of psychiatric genetics.

## 3. From a Polygenic Model to an Omnigenic Network Hypothesis of Psychiatric Disorders

A polygenic model views disorder risk from a multitude of common and rare variants as combinatorial contributions. More recent inferences from the polygenic model and advances in technology and biology have promoted the recognition of an omnigenic model, which views genetic architecture from a network perspective [22,23]. Networks are graphical models depicting interactions between nodes. From social networks to the World Wide Web, network models emphasize the structure of interconnections between nodes, which may have apparent commonalities across domains. In biological networks, nodes are biological entities such as genes, proteins, non-coding RNAs, and metabolites. Nodes are organized hierarchically and may be described in terms of their topology or interconnections, such as in scale-free architecture [24]. In a scale-free network, most genes have only a few connections with other genes, while a small proportion of genes have a high level of connectivity and are located at the center of the network. These small numbers of genes with high connectivity are referred to as ‘core genes’ or ‘hub genes.’ The remaining genes with low connectivity are referred to as ‘peripheral genes’ [25]. The relationships between entities are illustrated by edges connecting nodes, with the strength of interactions encoded as weights on edges. This view of genetic and signaling architecture permits not only relationally-based biological insights but also hypotheses regarding how certain structures are more vulnerable or robust to disorder.

In the omnigenic model, core genes and peripheral genes contribute differentially to the heritability of complex traits [22]. The small number of core genes usually plays a large regulatory role in the network, thus having large effect sizes. On the other hand, the majority of genes are peripheral genes, which account for most of the heritability as a whole, yet each displays a small effect size. Naturally, one can correspond to rare variants with core genes and common variants with peripheral genes (Figure 1c). While the polygenic model also allows for rare variants with large effect sizes, the omnigenic model provides additional insights into the underlying gene regulatory relationships responsible for pathogenesis. The assumption that rare variants are core genes in the network, which have larger effect sizes, account for a small percent of heritability, and are more phenotype-specific, is supported by existing studies [23,26,27]. Thus, we believe that the omnigenic model is superior in reflecting the underlying pathogenic mechanisms of complex psychiatric disorders.

The omnigenic network model calls for systems biology tools to make inferences about pathogenic mechanisms; however, variants alone are insufficient to construct disorder-related networks—additional molecular data that help establish or infer functional relationships are needed. Multiple levels of data from gene expression to protein interactions can be integrated to facilitate the construction of disorder-related networks (Figure 2). We describe these approaches in more detail below.

## 4. Connecting Disorder-Related Genetic Architecture to Network Models

Integrating and embedding multi-tissue, multi-omics data into network architectures offers unprecedented relational insights while anchored to physiologic contexts. Common omics data for network construction include genetics, transcriptomics, proteomics, and epigenomics [28]. Transcriptomic datasets derive from microarray, RNA-seq, and single-cell RNA-seq experiments and are the most used data type in network construction. mRNA and non-coding RNA expression levels can inform gene co-expression, regulation, and causality relationships. Epigenomics data such as histone modification, DNA methylation, non-coding RNA regulation, and open chromatin sites derived from methods such as CHIP-seq, ATAC-seq, Hi-C, and methyl-seq, highlight specific gene regulation profiles [29]. Proteomic data can reflect protein-protein physical interaction relationships based on assays such as the yeast double hybridization or co-regulation relationships through high-throughput methods such as protein chips. Across the omics domains, genetic and epigenetic variations contribute to gene expression regulation, which in turn affect protein levels and downstream protein-protein interactions and functions. All these within-datatype and between-datatype relations can be used in network construction.

As mentioned, the vast majority of heritability involves common variants that are often in non-coding areas, which cannot be directly mapped onto a gene network. To connect these loci with molecular networks in disease-relevant tissues, functional genomics serves as a bridge between genetics and other omics. Intermediate phenotype quantitative trait loci (iQTL) are mostly used for this purpose, which are genetic loci associated with specific quantitative traits such as gene expression or protein levels, which are intermediate traits between genetics and clinical phenotypes [30]. In terms of the quantitative trait associated, iQTLs include expression QTLs (eQTLs), splicing QTLs (sQTLs), histone modification QTLs (hQTLs), methylation QTLs (mQTLs), and protein QTLs (pQTLs) [31]. The most common type of iQTLs studied is eQTLs that define genetic loci that are associated with gene expression. eQTLs can be divided into cis-eQTLs and trans-eQTLs. Cis-eQTLs are adjacent genetic loci that cis-regulate the covariate gene, while trans-eQTLs are distant genetic loci that regulate genes remotely [32]. In a network, cis-eQTLs can help set the corresponding covariate genes as parent nodes. In contrast, trans-eQTLs can infer the covariate genes as child nodes. As such, eQTL information can also be directly incorporated into network construction. For instance, using eQTLs as input for Bayesian networks boosts causal inference and network performance [33]. In this way, one can locate disorder-related common variants in a network by examining the connection of their eQTL covariate genes and further identify hub genes related to common variants in the network.

As efforts devoting to large-scale omics profiling proceed, there has been an accumulation of databases of different data modalities that can be used for psychiatric research (Table 1). GWAS catalog, LD-hub, and PGC collect the summary statistics of genetic associations of diseases or phenotypes from many GWAS, including numerous psychiatric disorders [34,35]. The Genotype-Tissue Expression (GTEx) project profiled the genotype, transcriptome, eQTLs, and sQTLs across 54 tissues in a total of 948 donors, including 2642 samples from 13 brain regions [36]. The Encyclopedia of DNA Elements (ENCODE) profiles various transcriptional regulators and epigenomic factors across more than 150 tissues from 4920 samples, including 706 brain samples [37]. Another project focusing on transcriptional regulator profiling is the Functional Annotation of the Mouse/Mammalian Genome (FANTOM), which has released atlases of transcriptional regulatory networks, promoters, enhancers, lncRNAs, and miRNAs [38]. Apart from GTEx, an abundance of bulk tissue RNA-seq and single-cell RNA-seq datasets can be found at the Gene Expression Omnibus (GEO) [39]. Lastly, the Search Tool for the Retrieval of Interacting Genes/Proteins (STRING) curates and profiles protein interactions, with the latest version v11 including 24.6 million proteins from 5090 organisms [40]. All these data resources enable robust network construction integrating multi-tissue multi-omics datasets.

## 5. A Survey of Current and Potential Network Methods and Applications in Psychiatric Research

Networks commonly used in systems biology include gene regulatory networks, protein-protein interaction networks (PPI), literature-curated networks, and hybrid networks (Table 2). These network models depict the molecular relationships at both the cellular and intracellular level, each from a unique perspective. Gene regulatory networks focus on elucidating gene-gene interaction and regulatory relationships, organizing genes based on co-expression clusters or inferred causality and regulatory pairs. PPIs emphasize the physical interaction between proteins, combining protein interaction information from both experiments and computational predictions. Literature-curated networks capture potential gene or protein interactions by mining gene or protein co-occurrence from published research papers. Hybrid networks combine and integrate information from two or more different networks and present a comprehensive summary for a specific tissue [41].

Below we discuss each network method and its application in psychiatric disorders. We also highlight approaches that are not yet widely adopted in psychiatric research.

### 5.1. Gene Regulatory Networks

Three main kinds of gene regulatory networks are commonly adopted: gene co-expression networks, causal relationship networks, and regulator-target pair networks.

#### 5.1.1. Gene Co-expression Networks

Gene co-expression networks are correlation-based networks in which highly co-regulated genes are clustered into modules, illustrating the functional clustering of genes and pinpoint core genes based on connectivity. Commonly used methods to generate gene co-expression networks include WGCNA [42] and MEGENA [43]. The key differences between the two types of co-expression networks include the module size (large modules in WGCNA vs. more compact modules in MEGENA) and whether a gene can be in multiple modules (not allowed in WGCNA but allowed in MEGENA). WGCNA has been widely implemented in numerous studies and is one of the most adopted network methods in systems biology and psychiatric research. In contrast, MEGENA has not been broadly applied. In recent comparative applications between the WGCNA and MEGENA for non-psychiatric diseases, the complementary nature of the two methods is strongly supported [50,51]. It will be interesting to test MEGENA in psychiatric disorders in future studies.

Using WGCNA or MEGENA, one can identify modules associated with certain conditions based on the transcriptomic profiles in both cases and controls using the module-trait correlation analysis (Figure 3a). By annotating disorder-associated modules’ biological function, cell types and pathways responsible for pathogenesis can be elucidated. For example, Kapoor et al., examined bulk-tissue gene expression in the prefrontal cortex of subjects with alcohol use disorder as well as controls [52]. They applied WGCNA to the transcriptomic profiles and identified two modules that were significantly correlated with alcoholism. Further pathway analysis suggested that in subjects with alcoholism, there is a down-regulation of calcium signaling and nicotine response pathways in one module and an up-regulation of immune signaling pathways in another module.

Another common application of co-expression networks is first to construct co-expression networks based on the transcriptomic profiles from control subjects and then to examine the module enrichment level of genes affected in a specific disorder (Figure 3a), as exemplified in the following study. Parikshak et al. constructed WGCNA gene co-expression networks based on bulk tissue brain RNA-seq data from subjects representing the cortex of early developmental stages spanning post-conception, week 8 to one year after birth in the BrainSpan database [53,54]. ASD rare variant genes are enriched in hub genes of modules functioning in early transcriptional regulation and synaptic development. Spatially, ASD rare variant genes are enriched in superficial cortical layers and glutamatergic projection neurons of the cortex. These findings have been cross-validated experimentally by other studies using postmortem human brain samples from subjects with ASD, using both bulk-tissue and single nucleus transcriptomics [55,56]. A parallel paper by Willsey et al., also utilized the BrainSpan database and gene co-expression networks to identify pathways and cell types related to ASD [57]. Rather than using WGCNA, the authors constructed co-expression networks of high confidence of ASD ‘seed genes’ using the Pearson correlation coefficient to choose the top 20 best-correlated genes with a Pearson coefficient higher than 0.7 for each seed gene. Using this method, the authors elucidated an enrichment of ASD rare mutation genes in deep-layer glutamatergic projection neurons of the mid-fetal cortex, consistent with the Parikshak et al. findings.

Gene co-expression networks are powerful tools in determining co-regulatory relationships of genes involved in distinct functions and how these modules connect to psychiatric disorders. However, co-expression networks are not directional, and thus are unable to provide causal relationships between genes, an important aspect to retrieve upstream regulators. This limitation can be addressed by causal regulatory networks described in the following sections.

#### 5.1.2. Bayesian Networks (BNs)

BNs are directed acyclic graphs summarizing causal regulatory relationships between genes. BNs can be generated with transcriptomic data alone, but the incorporation of prior information capturing regulatory information can offer higher prediction accuracy for regulatory relationships [33,58]. cis-eQTLs, trans-eQTLs, and transcriptional factor-target pairs can be used as prior information for causal inference [32]. For instance, genes with cis-regulatory function and transcription factors are assigned as ‘parent nodes’ in BN, while genes under trans-regulation or target genes of transcription factors are ‘child nodes’. Arrows pointing from the parent nodes to child nodes indicate the inferred direction of causality.

Integrating genetics, transcriptomic, functional genomic inputs, and more, BN can capture causal regulatory relationships in a given tissue and can be used as a ‘roadmap’ in pinpointing key regulatory genes [58] (Figure 3b). There have been a few applications of BN in psychiatric research. Scarpa et al., leveraged a combination of a WGCNA gene co-expression network, transcription factor-target network, PPI, and BN to identify convergence and divergence of biological processes between sleep loss and depression [59]. The authors first measured the affective sleep patterns of 288 hybrid mice and their genotypes and transcriptional profile in the cortex, hippocampus, hypothalamus, and thalamus. WGCNA co-expression networks were constructed to identify trait-related modules in individual tissues, and BNs were constructed based on the transcriptomic profiles in conjunction with eQTLs derived from the genotype information and transcriptomics. Next, the authors examined the differentially expressed genes (DEGs) from a meta-analysis cohort of human major depressive disorder and from mouse sleep loss datasets. The DEGs of the human major depressive disorder and the mouse sleep loss model converged on a frontal cortex-derived module enriched in clock genes and immediate early genes (IEGs). Moreover, genes in this module displayed opposite directions of change in major depressive disorder subjects and sleep deficient mice, in line with the facts that many major depressive disorder patients manifest sleep issues and that antidepressants affect sleep. The authors then identified the key driver gene of this subnetwork by overlapping the non-directional co-expression modules on the directional BN to identify intramodular regulatory hub genes. An IEG *Arc* was found as a key driver gene of the clock/IEG network, which may link depression and sleep loss.

Protein-protein interaction and tissue-specific gene expression patterns have also been used to construct BNs, as in GIANT BNs [60]. GIANT BNs contain 31 central nervous system-related tissue-specific functional interaction networks, each constructed based on transcriptomic profiles, protein interaction information, and regulatory information curated from diverse experiments. Among those, the brain-specific BN was constructed with thousands of curated experiments by Krishnan et al., and was used to predict ASD risk genes and characterize their biological functions [61]. The authors applied this BN as an input to a machine-learning procedure, which was informed by text-mining co-occurrence of genes of high-confidence ASD associations. This approach revealed synaptic transmission, MAPK signaling, histone modification, and immune responses to be essential in affecting functions in ASD, which was cross-validated with the previous literature and experiments. Besides, this method was applied to prioritize driver genes in ASD-related CNVs, and the authors highlighted *PPP4C* and *MAZ* as potential top driver genes in the most common ASD-related CNV 16p11.2.

Although not yet widely adopted in psychiatric research, BN has been applied to study many other diseases such as Alzheimer’s disease [62], Type II diabetes [63], and non-alcoholic fatty liver disease [50]. BN is very powerful in identifying key driver genes in a biological process based on causality. For example, BN construction in RIMBANet uses Monte Carlo Markov Chain simulations to reconstruct 1000 networks starting with random seeds, and the final BN is a consensus network containing the most shared edges across all the reconstructions [59]. Although this method promotes causal inference, its disadvantages include high computational costs [29,41,44], the possibility of failing to find the optimal network structure [41], and a lack of feedback loops that misses an essential type of gene expression regulation [64].

#### 5.1.3. Regulator-Target Pair Networks

Dysregulation of transcription factors and non-coding RNAs have been indicated in psychiatric disorders [55,65,66,67]. A regulator-target pair subnetwork consisting of a gene expression regulator (such as a transcription factor or a non-coding RNA) and its downstream effect genes is termed a ‘regulon’. One can directly explore experiment-derived regulator-target pair networks from databases such as FANTOM and ENCODE [37,68]. An alternative method is to use the binding sites or transcriptomic information to infer targets of transcription factors or non-coding RNAs.

One piece of software that infers transcription factor regulons based on transcriptomic information is the Algorithm for the Reconstruction of Accurate Cellular Networks (ARACNe). Repunte-Canonigo et al., applied ARACNe to a rat model of alcoholism and identified *Nr3c1*, the gene encoding the glucocorticoid receptor, as a master regulator across many brain regions in alcohol-dependent rats [45,69]. The authors then performed an in vivo validation experiment by administering a glucocorticoid antagonist to the nucleus accumbens and ventral tegmental area of the alcohol-dependent rats and control animals. A significant decrease in alcohol consumption was observed in alcohol-dependent rats with the glucocorticoid antagonist to either of the two brain regions, while there was no effect in the alcohol non-dependent rats. Another method developed based on ARACNe, the Reconstruction of Transcriptional Regulatory Networks (RTN) [70], was applied by Pfaffenseller et al., to identify differentially expressed regulons in the prefrontal cortex of subjects with the bipolar affective disorder [71]. Five regulons (EGR3, TSC22D4, ILF2, YBX1, and MADD as regulators) were identified, with EGR3 showing the most robust significance in two independent human bipolar affective disorder datasets.

The Ingenuity Pathway Analysis (IPA), a commercial tool constructed based on a comprehensive curation of different networks from experimental datasets, text-mining literature, and other databases, also contains transcriptional factor and miRNA-target pair networks [72]. Using this tool, Bam et al., predicted many down-regulated miRNAs in PTSD to target *IFNG* and *IL-12*, which exhibit increased expression levels in PTSD patients [73]. They also predicted that the up-regulation of *hsa-miR-193a-5p* could decrease the high expression level of *IL-12*, which may help reduce the excessive inflammation response in PTSD patients.

Another software tool, TargetScan, predicts mRNA targets of miRNA based on conserved sequences in mRNAs [46]. Wu et al., leveraged a combination of the co-expression network and miRNA-target regulatory network to identify miRNA dysregulation in ASD [67]. The authors firstly identified differentially expressed miRNAs from ASD case-control brain samples and constructed WGCNA co-expression modules. They then used the TargetScan algorithm to identify mRNA targets of top differentially expressed miRNAs and hub miRNAs in ASD-related co-expression modules. The authors illustrated that ASD-related risk genes are enriched in miRNA targets and miRNA modules. One miRNA, *hsa-miR-21-3p*, targets neuronal/synaptic genes down-regulated in ASD, which may play an essential role in pathogenesis.

### 5.2. PPI Networks

In a PPI network, the nodes are proteins, and the edges depict the physical interaction relationship between proteins based on experimental datasets or computational simulation. The edges are undirected, and the weights of the edges indicate the reliability of the interaction. StringDB is the most commonly used PPI database, with the latest version v11 covering around 25 million proteins from 5099 organisms [40]. StringDB imports and integrates PPI information from other databases, including PINA, MINT, IntAct, DIP, BioGRID, HPRD, and MIPS/MPact. It also contains PPI inferred from text-mining, statistically significant co-occurring genes from the literature, and computationally predicted PPI based on criteria such as co-expression. Other integrated databases, including IPA and GeneMania, also contain PPI resources.

PPIs have been extensively used in psychiatric research to identify hub genes. Commonly used methods of identifying subnetworks and hubs include DAPPLE, DMS, MCODE, and PINA [74,75,76,77]. For example, Blizinsky et al., constructed a PPI network related to rare CNVs in schizophrenia using PINA2 [78]. MAPK3/ERK1 was identified as the most topologically important hub for the 16p11.2 network. The authors then performed an in vitro validation by applying an ERK signaling inhibitor to cultured primary neurons with 16p11.2 microduplication. This treatment successfully reversed the abnormality in these neurons’ dendritic arborization, indicating the critical role of MAPK3/ERK1 in maintaining normal neuronal morphology.

PPIs are also broadly used in combination with other networks, such as the co-expression network, to identify disorder-related networks. Gulsuner et al., profiled schizophrenia-related de novo mutations and leveraged a combination of the PPI network and co-expression network to examine the functional relevance of these de novo mutations and identify their enrichment in pathways and tissues [79]. By mapping de novo mutation genes onto the GeneMania physical interaction data set, the authors constructed an interconnected subnetwork enriched for schizophrenia de novo mutation genes, suggesting that the mutation genes are biologically interacting. The authors then constructed co-expression networks based on the BrainSpan data by calculating Pearson correlation coefficients across de novo mutation gene pairs. The most highly connected co-expression network of de novo mutation genes was derived from the fetal cortex. To further examine the interaction topology and gene characteristics, the authors merged the PPI network and co-expression network derived from the schizophrenia risk genes. This merged network contained genes in pathways related to neurogenesis and synaptic integrity, with most of the genes expressed high in early fetal development, low in childhood, and high again in early adulthood, which is in line with the onset of schizophrenia in early adulthood.

Since proteins interacting with each other may be co-regulated by the same upstream regulatory signal, they may also exist in the same co-expression network. Thus, genes that are both co-regulated and show protein interaction may be of higher relevance to a specific condition and should be prioritized for further study as driver genes (Figure 3b). Parikshak et al., identified two WGCNA co-expression modules enriched for ASD rare de novo variants [53] and further showed that these modules enriched for rare variants are also significantly enriched for protein interaction. The authors then intersected the WGCNA co-expression network hub genes with the PPI network hub genes. Many of the overlapping hub genes are known to harbor ASD-associated mutations and interact with other ASD-related genes, such as *TBR1*, *NFIA*, and *KDM6B*.

The comprehensive PPI databases provide abundant resources for reliable PPI networks. However, one main limitation of the PPI network is that it does not reflect causality or regulatory relationships. In addition, the PPI network is not tissue-specific and may bring in interactions that are not relevant to the disease tissues.

### 5.3. Literature-Based Networks

Literature-based networks are curated based on integrating general knowledge such as pathway/function annotation databases and text-mining existing experimental data and literature to capture genes contributing to specific biological functions. Although gene ontology terms can be constructed into a network in terms of the relatedness of biological pathways, genes in this type of network are not interconnected and do not reflect the topological properties of gene-gene relationships [80]. Thus, pathway/function annotation databases such as GO, KEGG, Reactome, and BioCarta are collections of sets of functionally-related genes rather than networks. However, pathway annotations can be used in conjunction with other molecular interaction information in network analysis.

Leveraging a knowledge-based and data-driven gene-phenotype likelihood network, Gilman et al., developed the NETwork-Based Analysis of Genetic Associations (NETBAG) [47]. The gene-phenotype likelihood network was first constructed by connecting all pairs of human genes. Then weights calculated based on the likelihood ratio were assigned to the edges based on the naive Bayesian integration of pathway annotations, protein-protein interaction information, shared gene or protein sequence or structure, and coevolutionary patterns. In this network, genes that are more likely to participate in the same phenotype have a high likelihood ratio weight. To predict genes affected by rare de novo CNVs in ASD using this network, the authors overlaid genes from ASD-related CNV regions and found the subnetwork with the highest enrichment p-value. They discovered that genes in these subnetworks mostly participate in synapse development, axon targeting, and neuron motility, which were cross-validated by later studies [53,55,56,57,61,81]. Gilman et al., also developed the NETBAG+ to incorporate more genetic variation, including GWAS loci and de novo single nucleotide variants. Further analysis suggested that the cortical interneurons, pyramidal neurons, and medium spiny neurons are the most impacted cell types in ASD [47]. The authors also applied the NETBAG+ to schizophrenia and identified related subnetworks and their functions [81]. Schizophrenia-related networks function mainly in axon guidance, neuronal cell mobility, synaptic function, and chromosomal remodeling, which largely coincides with ASD but with different mutations.

Similar to the NETBAG, Ward et al., constructed phenotypic-linkage networks (PLN) to identify nervous system gene sets related to the GWAS loci of mood instability [48]. The authors constructed a nervous-PLN using phenotypes specifically in the Nervous System mouse phenotype ontology (MPO) category, and integrated GO terms and pathway annotations with PPI and co-expression relationships, to derive semantic similarity scores [82]. Using this nervous-PLN, the authors found genes within loci related to mood instability to function in synapse transmission. Specifically, two candidate genes associated in the network are *HTR4* and *MCHR1*, encoding serotonin and melatonin receptors, respectively, and have been indicated in depression and schizophrenia.

### 5.4. Hybrid Networks

Apart from the individual usage of the networks mentioned above, constructing a hybrid network consisting of two or more kinds of networks is also a common practice. As different networks cover different aspects of gene interaction, a hybrid network can leverage the strengths and overcome the disadvantages and limitations of each network type. For example, combining PPIs with BNs integrates both causal gene regulatory information and protein physical interaction, which covers distinct aspects of gene interactions and promotes the identification of hub genes that play regulatory roles at the gene and/or protein levels.

Gazestani et al., constructed hybrid networks incorporating knowledge-based, functional, and experiment-derived co-expression networks to identify transcriptional perturbation patterns in leukocytes from ASD cases and control children [83]. The authors first generated a static network which combines information from (a) high-confidence physical and regulatory interactions from the Pathway Commons database, Reactome, and BioGRID; (b) co-expression network based on the transcriptome of the aforementioned case-controlled leukocyte samples; (c) functionally-related gene interactions from GeneMania, which includes PPI, co-expression, co-localization, pathways, and protein domain similarity information. Gene pairs in each diagnosis group (case or control) were retained to generate diagnosis-specific networks. By comparing the control-specific network and ASD-specific network, the authors discovered that the ASD network was enriched with ASD rare mutation genes, as well as their regulatory targets and regulators. RAS–ERK, PI3K–AKT, and WNT–β-catenin signaling pathways were enriched in ASD-specific networks, and ASD rare mutations perturbed the network through these pathways.

In addition to constructing literature-curated networks or hybrid networks from scratch, there are numerous existing hybrid network resources available. Huang et al., benchmarked 21 popular human gene or protein networks, including StringDB, GeneMania, and GIANT, based on a disease gene set recovery test and found that networks with a larger size have better performance in retrieving known disease genes [49]. They then assembled an integrative network that requires edges to be present in at least two networks, called Parsimonious Composite Network (PCNet), which is smaller in size but has the best performance. Although psychiatric disorders were not explicitly tested, this study provides a new network resource and a guideline to choose hybrid network resources.

### 5.5. Cross Disorder Network Applications

Due to the genetic correlation and comorbidity across psychiatric disorders, studying pathogenesis mechanisms across disorders may yield therapeutic targets for several disorders or specific endophenotypes. A recent study by Gandal et al., leveraged WGCNA gene co-expression networks constructed based on case-control human brain transcriptomic profiles across five psychiatric disorders: schizophrenia, major depressive disorder, bipolar affective disorder, ASD, and alcohol use disorder [84]. In addition to the distinct transcriptomic disturbance in each disorder, the authors identified a shared component of transcriptional dysregulation across all five disorders related to the degree of polygenic overlap. Their results agree with previous findings supporting that a shared causal genetic component underlies all psychiatric disorders [85,86]. Further, the authors identified shared and unique modules across these disorders. In ASD, bipolar affective disorder, and schizophrenia, an astrocyte module with the annotation of glial differentiation is up-regulated; several modules associated with neuronal and mitochondria function are down-regulated in these disorders. A microglia module is uniquely up-regulated in ASD, which is confirmed in another study by Gandal et al., where modules related to microglia and the interferon response are significantly up-regulated in ASD but down-regulated in bipolar affective disorder and schizophrenia. There is also a shared upregulation of the NF-kB pathway across these three disorders [87].

The conclusions from the Gandal et al., studies are consistent with previous pathway analyses based on genetic risks and studied leveraging of a similar network approach. Using pathway analysis, a study by PGC found that common genetic risks of schizophrenia, major depressive disorder, and bipolar affective disorder converge on neural, immune, and histone modification pathways [86]. Kim et al., constructed WGCNA co-expression networks to identify shared modules across schizophrenia, major depressive disorder, and bipolar affective disorder, which were enriched with GABAergic markers, synaptic proteins, and immune functions. Interestingly, the specific genes in the immune function-related modules showed no overlap across all three disorders, indicating possible differential responses in the immune system [88].

### 5.6. Network Applications on Treatment Response

Besides network applications in studying pathogenesis, networks are also powerful tools to identify driver genes in treatment response. Although many psychopharmacological drugs are available and have been applied for more than five decades, their mechanisms of action remain mostly elusive. Moreover, drug responses in individuals differ tremendously. Elucidating the mechanisms of action and identifying biomarkers that predict individual responses to drugs can greatly aid precision medicine in psychiatry and the development of novel therapeutics.

Lithium (Li) is a first-line mood stabilizer for bipolar disorder, although its mode of action is not fully elucidated. To reveal the mechanism of Li in treating bipolar disorder and the differential responses across patients, Breen et al., characterized the transcriptomic profiles of subject-derived lymphoblastoid cell lines from Li responders and non-responders [89]. WGCNA gene co-expression modules suggested that Li treatment correlates to an upregulated immune response, apoptosis, and protein processing in the endoplasmic reticulum, and a down-regulated ribosome pathway, translation initiation, and phosphatidylserine metabolism. They further discovered that DEGs between Li responders and non-responders are enriched in cell cycle processes and nucleotide excision repair pathways. To identify psychopharmacological drugs with a similar transcriptomic signature to Li for bipolar disorder patients, the authors then queried the DEGs from Li treatment against DSigDB, a database of drug/compound-activated gene expression signatures [90]. Clonidine, an alpha2-adrenoceptor agonist, exhibited a drug-gene signature most reminiscent of the Li signature, thus has potential for bipolar disorder treatment. Besides DSigDB, CLUE and Metacore (commercial) also contain drug-gene transcriptomic signatures applicable for studying drug action mechanisms and drug repurposing.

### 5.7. Summary of New Insights Obtained from Network Studies of Psychiatric Disorders

Network methodologies provide us with a perspective beyond the identified disorder-associated variants. For instance, candidate genes in the network which show high connectivity to many previously identified genes can be prioritized as hub genes or driver genes. Due to their high connectivity, these hub genes may act as the converging points of disorder-related variants and pathways, making them potential targets for therapeutics even if they have not been implicated by genetic evidence yet. Besides revealing hidden novel genes in disorder etiology, network methodologies can elucidate regulatory relationships and coherent biological functions between disorder genes. For example, co-expression networks can identify covariation across modules of genes and differentially regulated modules, while one may fail to identify significant differentially expressed genes.

Many fruitful findings have been made through network methods, which are summarized in Table 3. Among the six psychiatric disorders discussed in this review, ASD and schizophrenia are the most studied, and independent studies have yielded consistent results. Existing studies exploiting network methods mostly focus on identifying candidate genes and pathways based on transcriptomic studies (Figure 4). Almost every study on ASD has indicated the pathogenic role of immune and synaptic functions in ASD pathogenesis. Other crucial biological processes revealed for ASD include chromatin and transcription regulation, early embryonic development, axon guidance, extracellular matrix, and MAPK signaling [47,50,53,57,83,91,92,93,94,95]. Besides the processes affected in general ASD cases, Luo et al., combined electronic health records with genomic and transcriptomic data and identified an ASD subtype with dyslipidemia [96]. In addition, the various studies also implicated key cell types related to ASD, including mid-fetal deep layer cortical projection neurons, superficial cortical layers neurons, cortical interneurons, medium spiny neurons, and microglia [55,57,81,97].

Schizophrenia also engages the immune system, synaptic functions, and neurodevelopmental processes, which fall into the same pathway category as ASD [111,113]. However, schizophrenia exhibits differential alterations in these pathways. Apart from the differential immune response discussed in Section 5.5 (upregulated microglia and immune activities in ASD and down in schizophrenia), the candidate genes from de novo CNVs of ASD and schizophrenia showed opposite directions in their biological functions. Most of the schizophrenia candidate genes are associated with synaptic pruning and decreased dendritic spines, while ASD candidate genes are associated with increased dendritic spines, which were also observed in postmortem brain analyses [111]. Besides, genes related to schizophrenia de novo mutations mostly show a characteristic expression pattern: high in the fetal stage, low in childhood, and high again in early adulthood [79], while ASD de novo mutations exhibit high expression in fetal and early postnatal development [53,114]. Another independent study on the 22q11.2 deletion associated with schizophrenia identified two hub genes that are expressed during embryonic brain development and adolescence, respectively [115]. This pattern coincides with the typical onset time of schizophrenia, which is around early adulthood. Schizophrenia-related genes have also been shown to fail to decrease naturally as control subjects do [107].

The polygenic component of bipolar affective disorder overlaps with schizophrenia significantly, but network application in bipolar affective disorder is very limited. The role of postsynaptic density in bipolar affective disorder pathogenesis has been indicated by independent studies [101,116]. Hub genes such as *MAP4* and *ILF2* were also suggested, but due to fewer study numbers and a lack of validation, a consensus cannot be reached [71,100,102].

Major depressive disorder and PTSD are stress-related disorders that share neuronal and immune dysregulations based on network studies [104,117,118,119]. PTSD has been shown to engage immune processes more prominently than major depressive disorder. Dysfunction of multiple immune processes, including innate immunity, interferon responses, cytokine receptor interaction, and glucocorticoid receptor activity, has been implicated [106,117,120,121,122,123]. Unlike PTSD, for which all network studies identified immune dysregulation, the mechanism behind major depressive disorder seems less coherent across studies similar to the case of bipolar affective disorder. In a study using a mouse model to identify hub genes related to depression susceptibility, several key drivers including *Dkkl1*, *Neurod2*, and *Sdk1* were validated in vivo, indicating the reliability of network predictions and implicating synaptic transmission, cell-cell signaling, and oxidative phosphorylation pathways in depression pathogenesis [124].

Lastly, alcohol use disorder is a disorder combining features of addiction and neurotoxicity. Network studies of alcoholism have revealed processes related to mitochondrial dysfunction, synaptic transmission, neuroplasticity, calcium signaling, and immune functions [52,98,99,125,126]. One hub gene, *Nr3c1*, predicted by transcription factor network was validated in vivo in an alcohol-dependent mouse model [69]. More studies are needed to reveal the underlying mechanisms behind alcoholism.

## 6. Conclusions and Future Directions

In summary, we have introduced and illustrated the main network approaches, their strengths and limitations, and how they can complement one another by highlighting relevant studies. Despite the aforementioned recent discoveries, network applications in psychiatric research are still in their infancy. Networks such as the WGCNA co-expression network and PPI have been extensively applied, while other networks such as BN are rarely adopted despite indications as being powerful tools in other research fields. Integrating these network approaches may reveal hidden pathogenic mechanisms by capturing underappreciated information from the data. We recommend the adoption of diverse types of network approaches in each study to derive comprehensive molecular insights.

Besides leveraging complementary network methods, another future direction would be to have an integrated framework followed by the field to apply a set of benchmarked and well-performing networking methodologies systematically. Such a framework would eliminate technical bias caused by different methods to enable systematic comparison of psychiatric disorders at a network level.

Benchmarked and standardized network methodologies are applicable regardless of disorder types. However, in order to better elucidate trait-specific biology, we recommend careful collection of multi-omic data types that reflect the unique aspects of a certain disorder, including specific causal factors (e.g., genetic versus environmental) and the corresponding omics data types (e.g., genetic variants for genetic causes; epigenetic alterations for environmental causes), and related brain regions and circuits at the relevant developmental stages. In addition to collecting relevant types of data, tissue heterogeneity needs to be addressed as a future direction. As a highly complex organ, the brain consists of numerous subregions and nuclei, each containing various cell subtypes. Previous network application studies in psychiatric disorders were mostly performed at the brain region level using bulk tissue transcriptome. Obviously, with the advancement of single cell omics technologies, exploring cell-level networks becomes an urgent need. The abundance of single-cell transcriptomic datasets enables researchers to further dissect the pathogenic mechanisms of psychiatric disorders at an increasing granularity at the cell type or subtype level. Thus, it is possible to identify cell subtypes related to a specific condition and pinpoint key driver genes in different cell subtypes.

Appealing as it is, network methods for single-cell datasets are still limited. Due to the challenge of data sparsity, methods applicable to bulk tissue transcriptomics do not perform well on single-cell datasets [127]. However, a few single-cell network methods have been successfully applied widely in studying cell type diversity and non-psychiatric conditions, including in a ligand-receptor binding network, single-cell gene regulatory network, and single-cell co-expression network. The ligand-receptor binding network is a PPI network emphasizing intracellular interactions. By looking at the ligand and receptor pairs expressed in cell types, we can identify interacting cell types utilizing autocrine, paracrine, and endocrine signaling. For example, CellPhoneDB and iTALK are two standard tools to calculate cell-cell interaction scores [128,129]; SCENIC uses transcription factor information and single cell transcriptome data to identify regulons at a cell-type-specific level [130]; scLink infers gene co-expression networks from a sparse gene expression matrix [131]; CytoTalk aims to construct both within cell-type and between cell-type signaling networks [132]. Benchmarking and applying these methods would bring mechanistic research of psychiatric disorders to a finer granularity from the brain region level to the cell subtype level.

Besides using single cell omics as a resource providing pathophysiological insights with increasing granularity, single-cell and bulk tissue transcriptomic profiling can also be applied as an approach to validating in silico predictions. Experimental validations have been limited despite the current progress of in silico findings in key drivers and pathways. More experimental validations should be performed to facilitate the transition from in silico predictions to the bench and eventually to the bedside. Current experimental validation methods include RT-PCR and transcriptomics for evaluating possible expression alteration of the key drivers from samples with the disorder [51,52,60]; human genetic studies for identifying risk genes [7,11,14,15]; and in vitro and in vivo experiments using an appropriate animal model or human subject-derived material for validating the molecular, cellular and behavioral phenotypes upon disrupting key driver expression [78,110,133,134] (Figure 3c).

In conclusion, approaching psychiatric genetics from a network perspective enables researchers to identify the converging pathways in genetic architecture and leverage the abundance of the omics databases to yield a better understanding of the pathophysiology and predictions for therapeutic targets. With this review, we hope to provide a systematic overview of network methodologies, previous network applications, and their findings in psychiatric research. Much remains to be explored—including adopting network approaches from other fields, standardizing a benchmarked and integrated framework, developing single-cell network construction methods, and performing corresponding experimental validations.

## Figures and Tables

**Figure 1 genes-12-01101-f001:**
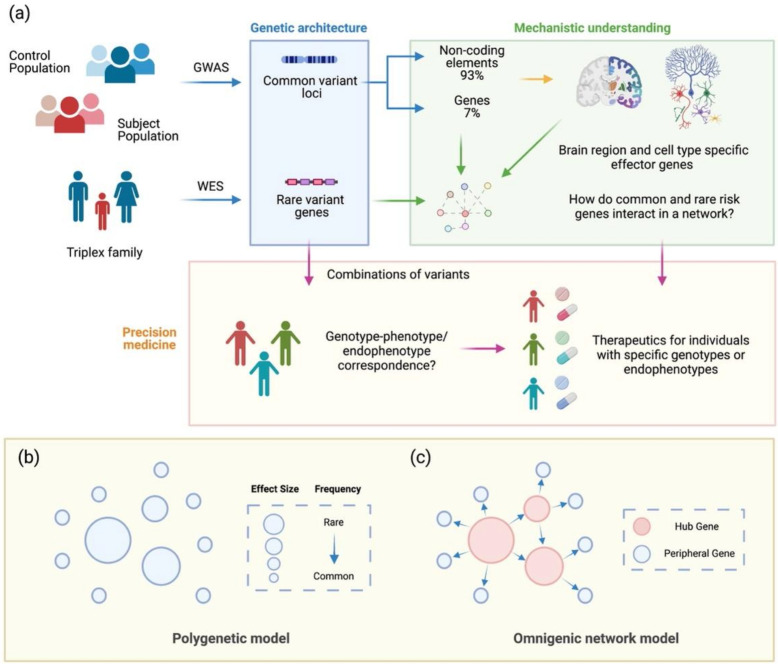
Pursuing a network understanding of psychiatric disorders’ genetic architectures to advance precision medicine. (**a**) With the increasing abundance of genome-wide association studies (GWAS) and whole exome sequencing (WES) studies, genetic data for psychiatric disorders are increasingly comprehensive. However, we still lack a mechanistic understanding of the genetic architecture in the pathogenesis of different disorders and symptoms. Establishing such an understanding systematically could enable the development of therapies for subgroups of patients or even on a personalized basis. Network modeling of gene interactions provides a powerful tool to dissect risk-gene relationships and pathways affected. The polygenic model and the omnigenic model are proposed for psychiatric disorders. In the polygenetic model (**b**), a certain trait is determined by a combination of multiple variants with different effect sizes. Common variants have a high population frequency and small effect sizes, while a small number of rare variants have a low population frequency but large effect sizes. In the omnigenic network model (**c**), the regulatory relationships between variants are depicted by the network. A small number of hub genes regulates the majority of peripheral genes. Rare variants likely reside in hub genes and common variants in peripheral genes.

**Figure 2 genes-12-01101-f002:**
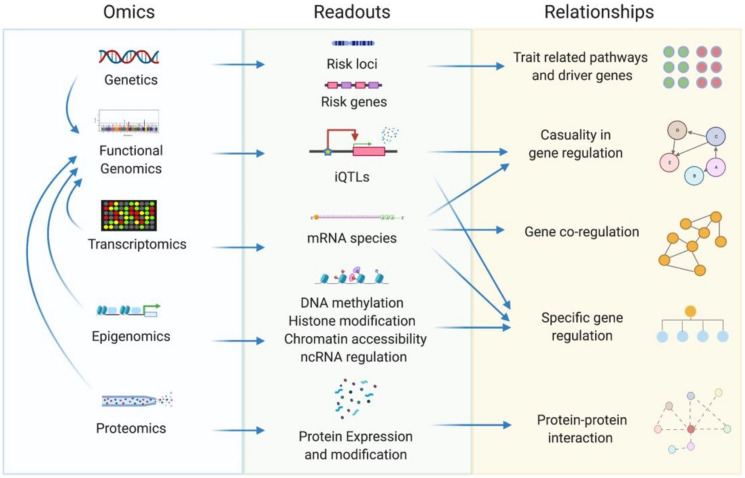
Extracting tissue and cell type-specific gene interaction relationships from multi-omics data. Genetics, functional genomics, transcriptomics, epigenomics, and proteomics are the most commonly used omics tools in obtaining gene interaction relationships. Genetic readouts can be used to infer trait-related pathways and driver genes, while readouts from other omics tools indicate gene regulatory relationships or protein-protein interactions. Particularly, intermediate phenotype QTLs (iQTLs) such as expression QTLs (eQTLs) or protein QTLs (pQTLs) from functional genomics data act as a bridge linking genetics and other omics by tissue-specific loci-gene regulatory relationship, thus enabling the interpretation of common variant loci in the non-coding areas of the genome.

**Figure 3 genes-12-01101-f003:**
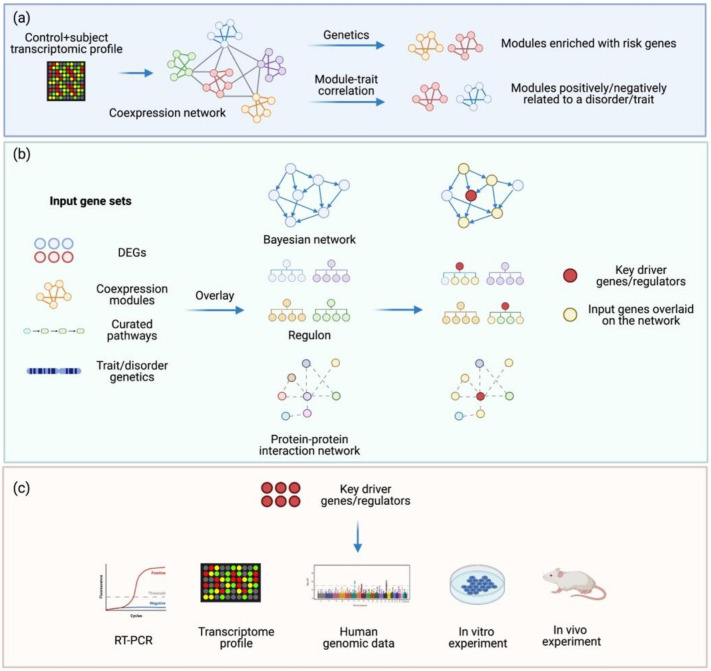
Using networks to identify disorder-related networks and key driver genes. (**a**) The pipeline to identify disorder-related co-expression modules. A co-expression network is generated from the transcriptomic profiles of a subject with a specific disorder and the corresponding controls using methods such as WGCNA. By calculating the enrichment level of disorder-related risk genes in each module, modules enriched with risk genes can be identified. Alternatively, modules positively or negatively correlated with the disorder can be identified by doing a module-trait correlation analysis. Downstream annotation of these modules’ biological functions will reflect pathways affected in the disorder. However, co-expression networks are unable to capture direct, causal relationships, which can be supplemented by Bayesian networks and regulator-target pair networks. (**b**) Using networks as a ‘road map’ to identify key driver genes of a specific disorder. Bayesian networks (BNs), regulons from regulator-target pair networks, and PPI networks depict causality, regulation, or direct physical interactions, respectively, and can be used as network models summarizing regulatory or direct gene-gene interactions in a certain tissue. By overlaying disorder-related gene sets, e.g., differentially expressed genes (DEGs), disorder-correlated co-expression modules, related pathways, and risk genes, one can pinpoint potential key drivers based on the topology of the networks. (**c**) A summary of key driver validation approaches. RT-PCR and transcriptomics can evaluate possible expression alterations of the key drivers from samples with the disorder. Key drivers may be also validated if they are identified as risk genes by human genetic studies. In vitro and in vivo experiments using appropriate cell or animal models may help to validate the molecular, cellular, and behavioral phenotypes upon disrupting key driver expression or activities.

**Figure 4 genes-12-01101-f004:**
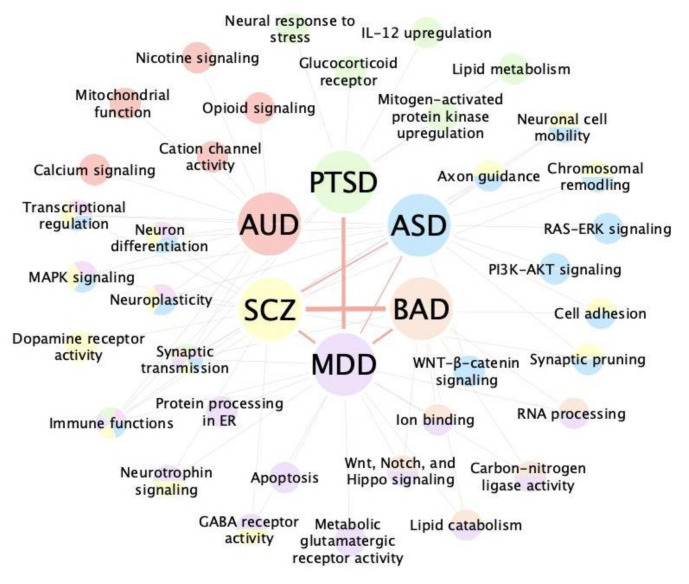
Comparing shared and distinct pathways across selected psychiatric disorders. Top pathways related to selected psychiatric disorders are shown in the disorder-pathway network. Red lines connecting disorders depict the correlation level between disorders [112]. Synaptic transmission-related processes are shared among all six disorders and immune functions are indicated in all disorders except BAD. Pathways including MAPK signaling and transcriptional regulation are shared among the AUD, ASD, and SCZ. Each disorder and its associated pathways are annotated with the same node color. Shared pathways between disorders are indicated with multi-color nodes. ASD: autism spectrum disorder; AUD: alcohol use disorder; BAD: bipolar affective disorder; MDD: major depressive disorder; PTSD: post-traumatic stress disorder; SCZ: schizophrenia.

**Table 1 genes-12-01101-t001:** Publicly available repositories of multi-tissue multi-omics data related to psychiatric research.

Omics	Database	Description	URL	Usage in Network Applications
Genetics	GWAS catalog	Collections of the GWAS summary statistics files	https://www.ebi.ac.uk/gwas/ (accessed on 11 January 2021)	Find trait-related genes, pathways, and subnetworks
	LD-hub		http://ldsc.broadinstitute.org/ (accessed on 11 January 2021)	
	PGC		https://www.med.unc.edu/pgc/ (accessed on 11 January 2021)	
Genomics/Functional genomics/Transcriptomics	GTEx	Genotype, transcriptome, eQTLs, and sQTLs profiles across 13 brain regions from 948 donors and 2642 samples	https://www.gtexportal.org/home/ (accessed on 11 January 2021)	“Building bricks” for gene regulatory networks
	GEO	A repository for various data types including genotypes, bulk tissue RNA-seq and single-cell RNA-seq datasets	https://www.ncbi.nlm.nih.gov/geo/ (accessed on 11 January 2021)	
	PsychENCODE	A repository specifically for neuropsychiatric disorders including RNA-seq datasets, SNP genotypes, epigenomic datasets and gene regulatory networks	http://resource.psychencode.org/ (accessed on 11 January 2021) https://www.synapse.org/#!Synapse:syn4921369/wiki/235539 (accessed on 11 January 2021)	
	BrainSpan	Transcriptional profiles of 16 cortical and subcortical regions with a temporal coverage across pre- and post-natal development in both males and females	http://www.brainspan.org/static/download.html (accessed on 11 January 2021)	
Epigenomics	ENCODE	Transcriptional regulator and epigenomic factor profiles from 706 brain samples	https://www.encodeproject.org/ (accessed on 11 January 2021)	Provide regulator-target pair information
	FANTOM	Atlases of transcriptional regulatory networks, promoters, enhancers, lncRNAs, and miRNAs	https://fantom.gsc.riken.jp/ (accessed on 11 January 2021)	
Proteomics	STRING DB	Curated protein interactions including 24.6 million proteins from 5090 organisms	https://string-db.org/ (accessed on 11 January 2021)	Provide protein-protein interaction information

**Table 2 genes-12-01101-t002:** Major networks used in psychiatric disorder research.

Networks		Relationship Captured	Disadvantages	Example Construction Methods
Gene regulatory network	Co-expression network	Covariation and co-regulation among gene clusters	- Not directional - No causal relationships	WGCNA [42]; MEGENA [43]
	Bayesian network	Causality of regulation between gene pairs	- High computational cost - Lack of feedback loops - Possibility of failing to find the optimal network structure	RIMBANet [44]
	Regulator-target pair network	Specific regulation of certain transcriptional factors/non-coding RNAs	- Only captures certain types of regulator relationships	From database (FANTOM) [38]; ARACNe [45]; TargetScan [46]
Protein-protein interaction network		Physical interaction affinity between pairs of proteins	- Cannot reflect causality or regulator relationships - Current PPI datasets are not tissue-specific	From database (STRINGDB) [40]
Literature-based network	Background likelihood network	Possibility of gene pairs participating in a similar genetic phenotype	- Limited by the current level of knowledge	Gilman et al., 2011 [47]
	Phenotypic-linkage network	Gene clusters related with disease-related phenotypes curated from the literature		Ward et al., 2020 [48]
Hybrid network		General gene-gene interactions, PPIs, and literature co-occurrence		Use premade networks (e.g., PCNet) [49]; Custom script

**Table 3 genes-12-01101-t003:** Key findings based on network applications in selected psychiatric disorders.

Disorder	Networks	Key Findings	Ref.
ASD	Co-expression network	Synapse and immune response-related modules are affected in frontal and temporal cortex from ASD subjects; ASD rare variants affects early transcriptional regulation and synaptic development pathways and are enriched in superficial cortical layers and glutamatergic projection neurons in developing and adult human cortex.	Voineagu et al. [91]; Parikshak et al. [53]
	Protein-protein interaction network	ASD rare variant related protein interactions are enriched in synaptic transmission, cell junction, TGFβ pathway, neurodegeneration, and transcriptional regulation.	de Rubies et al. [93]; Sanders et al. [92]
	Bayesian network	Synaptic transmission, MAPK signaling, histone modification, and immune response are the top affected functions in predicted ASD risk genes using a brain-specific network.	Krishnan et al. [61]
	Literature-based network	ASD rare variant genes form a network related to synapse development, axon targeting, and neuron motility; Genes in ASD rare variant and single nucleotide variants informed network are expressed at the highest level in cortical interneurons, pyramidal neurons, and the medium spiny neurons of the striatum.	Gilman et al. [47]; Chang et al. [81]
	Hybrid network	The ASD network constructed with the peripheral blood transcriptome in children with ASD was enriched for ASD rare mutation genes, as well as their regulatory targets and regulators. RAS–ERK, PI3K–AKT, and WNT–β-catenin signaling pathways are enriched in ASD-specific networks.	Gazestani et al. [83]
AUD	Co-expression network	In prefrontal cortex samples from human AUD subjects, a module functioning in calcium signaling, nicotine response and opioid signaling are down-regulated in AUD, while another module functioning in immune signaling are up-regulated in AUD; In nucleus accumbens samples from human AUD subjects, two neuronal modules enriched for genes in oxidative phosphorylation, mitochondrial dysfunction, and MAPK signaling pathways are down-regulated in AUD, while four immune-related modules enriched for astrocyte and microglia markers are up-regulated in AUD.	Kapoor et al. [52]; Mamdani et al. [98]
	Transcription factor/miRNA regulons	Pathways related to synaptic processes and neuroplasticity are disrupted in a rat AUD model; *Nr3c1* acts as a master regulator in multiple brain regions in alcohol-dependent rats.	Tapocik et al., 2013 [99]; Repunte-Canonigo et al., 2015 [69]
BAD	Co-expression network	BAD common variants are enriched in the hippocampus and amygdala across developmental stages. In dorsolateral frontal cortex samples from human BAD subjects, modules enriched for genes related to postsynaptic density, RNA processing, and carbon-nitrogen ligase activity are downregulated, while modules enriched for genes related to ion binding and lipid catabolism are upregulated.	Xiang et al. [100]; Akula et al. [101]
	Transcription factor regulons	EGR3, TSC22D4, ILF2, YBX1 and MADD are predicted as master regulators in human prefrontal cortex with BAD.	Pfaffenseller et al. [71]
	Protein-protein interaction network	*CDH4, MTA2*, *RBBP4*, and *HDAC2* are the core genes predicted by PPI analysis, involved in early brain development regulation. *HP* and *PC* are related to BAD de novo mutations; *MAP4*, *WDHD1*, *EIF4E* and *STRN* are related to the BAD common variant loci.	Xiang et al. [100]; Toma et al. [102]
MDD	Co-expression network	*CCND3*, *TXND5*, *TRI26* are the driver genes for cognitive dysfunction in MDD, validated by plasma protein level in MDD subjects; Immune response and protein processing in the ER are disrupted in older adults with recurrent MDD	Schubert et. al. [103]; Ciobanu et al. [104]
	Protein-protein interaction network	The *ATP5G1* gene is associated with the pathogenesis of MDD	Zeng et al. [105]
PTSD	Co-expression network	Differential responses to PTSD are observed in correlated modules constructed from the peripheral blood transcriptome of PTSD subjects. In men, an IL-12 signaling module is upregulated; In women, a module related to lipid metabolism and mitogen-activated protein kinase is upregulated. Cytokine, innate immune, and type I interferon-related modules are shared between sexes.	Breen et. al. [106]
	miRNA regulons	Downregulated miRNAs in peripheral blood transcriptome of PTSD subjects are predicted to target *IFNG* and *IL-12*.	Bam et al. [73]
SCZ	Co-expression network	Genes related to central nervous system development failed to attenuate with age in SCZ subjects; Synaptic protein co-expression was significantly decreased in the auditory cortex of SCZ subjects; SCZ common variants are enriched in negative co-expression genes of C4A	Torkamani et al. [107]; MacDonald et. al. [108]; Kim et. al. [109]
	Transcription factor regulons	TCF4 is a master regulator identified from postmortem dorsolateral prefrontal cortex of SCZ subjects and cultured olfactory neuroepithelium	Torshizi et. al. [110]
	Protein-protein interaction network	MAPK3/ERK1 is the top hub for the 16p11.2 microduplication network	Blizinsky et. al. [78]
	Literature-based network	SCZ rare variant-derived network genes function mainly in axon guidance, neuronal cell mobility, synaptic function, and chromosomal remodeling, and are highly expressed in the brain during prenatal development.	Gilman et. al. [111]

ASD: autism spectrum disorder; AUD: alcohol use disorder; BAD: bipolar affective disorder; MDD: major depressive disorder; PTSD: post-traumatic stress disorder; SCZ: schizophrenia.

## Data Availability

Not applicable.

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
