# Peer review of "Unveiling the Pathogenesis of Psychiatric Disorders Using Network Models"

_genes, 2021, doi:10.3390/genes12071101_

Round 1

Reviewer 1 Report

This is a very nice systematic review regarding the most recent advances in hypothesis and methodologies in genetic research about 5 major psychiatric disorders, i.e., by using various network approaches under an omnigenic model. The review has provided quite thorough and detailed available resources and some convincing examples in each of these five major psychiatric disorders. My only comments:

  1. Better statement about the difference between polygenic and omnigenic models, the latter should replace the former one? 
  2. It could be more clearer if in Table 2 and Figure 3, plus the corresponding text part, the authors could illustrate how each type of network analysis could supplement or compliment each other to determine the trait-specific pathways and mechanisms? What will be the trait-specific determining or critical factors or what types of data and network will provide trait-specific information?  Or only combined various network analyses will do so?       

Author Response

We deeply appreciate the reviewer’s positive feedback and constructive suggestions. Below please find our point-by-point responses. Please see the attachment for attached Table 2. 

Reviewer 1:

This is a very nice systematic review regarding the most recent advances in hypothesis and methodologies in genetic research about 5 major psychiatric disorders, i.e., by using various network approaches under an omnigenic model. The review has provided quite thorough and detailed available resources and some convincing examples in each of these five major psychiatric disorders. My only comments:

  1. Better statement about the difference between polygenic and omnigenic models, the latter should replace the former one? 

Response:

We appreciate the suggestion and have made this point clear in section 3 (line 175-177) by adding the following statement:

“Thus, we believe that the omnigenic model is superior in reflecting the underlying pathogenic mechanisms of complex psychiatric disorders.”

We have also made corresponding changes in the abstract (line 19-21):

“There has been accumulating evidence suggesting that the genetics of complex disorders can be viewed through an omnigenic lens, which involves contextualizing genes in highly interconnected networks.”

  1. A) It could be clearer if in Table 2 and Figure 3, plus the corresponding text part, the authors could illustrate how each type of network analysis could supplement or complement each other to determine the trait-specific pathways and mechanisms?

Response:

We appreciate the suggestion and have modified Table 2 and Figure 3 text to better illustrate the strengths, limitations, and the complementary nature of the network methods. In Table 2 (attached in Page 4), we have added a new column to illustrate the limitations of each network method which can be complemented by other methods. In Figure 3 legend (lines 308-311), we clarified how different methods can complement one another as follows:

“(a)...However, coexpression networks are unable to capture directed, causal relationships, which can be supplemented by Bayesian networks and regulator-target pair networks. (b) Using networks as a ‘road map’ to identify key driver genes of a specific disorder. Bayesian networks, regulons from regulator-target pair networks, and PPI networks depict causality, regulation or direct physical interactions, respectively, and can be used as network models summarizing regulatory or direct gene-gene interactions in a certain tissue.”

  1. B) What will be the trait-specific determining or critical factors or what types of data and network will provide trait-specific information?  Or only combined various network analyses will do so?  

Response:

We believe to derive trait-specific information, it is more important to consider the types of data to be collected rather than a specific network methodology. Collecting multiomics data from disorder-relevant brain regions that best reflect factors that determine specific trait etiology will enables the various network methodologies to better shed light on the underlying trait-specific biology. A combination of various network analyses provides a more comprehensive view of molecular interactions by capturing different types of relationships. We have elaborated these points in the conclusion section as follows:

Lines 719-728: “In summary, we have introduced and illustrated main network approaches, their strengths and limitations, and how they can complement one another by highlighting relevant studies...... We recommend adoption of diverse types of network approaches in each study to derive comprehensive molecular insights.”

Lines 734-740: “Benchmarked and standardized network methodologies are applicable regardless of disorder types. However, in order to better elucidate trait-specific biology, we recommend careful collection of multiomics data types that reflect the unique aspects of a certain disorder, including specific causal factors (e.g., genetic versus environmental) and the corresponding omics data types (e.g., genetic variants for genetic causes; epigenetic alterations for environmental causes), related brain regions and circuits.

Reviewer 2 Report

It has been a pleasure to revise this review paper. The submission is very well structured and the quality of the writing and figures is outstanding. This will be a reference review for network analysis of multiomic datasets.

Minor comment

On page 4, section 3 from a polygenic model to an omnigenic network, I would like to suggest a short paragraph describing the extreme phenotype strategy in quantitative traits to identify rare variants with large effect size in complex disorders.

Suggested additional references

  1. Cai N, Bigdeli TB, Kretzschmar W, et al. Sparse whole-genome sequencing identifies two loci for major depressive disorder. Nature. 2015;523(7562):588-591. doi:10.1038/nature14659.
  2. Amanat S, Requena T, Lopez-Escamez JA. A Systematic Review of Extreme Phenotype Strategies to Search for Rare Variants in Genetic Studies of Complex Disorders. Genes (Basel). 2020;11(9):987. doi:10.3390/genes11090987

Author Response

We deeply appreciate the reviewer’s positive feedback and constructive suggestions. Below please find our point-by-point responses. 

Response:

We appreciate the positive comments and the insightful suggestion. We have added the suggested references and modified the text (lines 95-103) to describe the extreme phenotype strategy as follows:

One efficient way to characterize the genetic architecture of complex diseases is to search for protein-encoding rare mutations in singletons or multiplex families with extreme phenotypes, which include an early onset, more severe symptoms, or fast progression of diseases [13]. Human genetics studies of extreme psychiatric phenotypes and rare syndromes involving psychiatric symptoms have revealed numerous rare variants in psychiatric disorders. These rare variants include copy number variations (CNVs) and protein-altering point mutations; particularly for schizophrenia and autism spectrum disorder, 159 and 136 rare variants have been identified, respectively [7,14-17].
